# StreamBP: Memory-Efficient Exact Backpropagation for Long Sequence Training of LLMs

**Qijun Luo**[1]    **Mengqi Li**[1]    **Lei Zhao**[2]    **Xiao Li**[1*]

[1]The Chinese University of Hong Kong, Shenzhen

[2]Shanghai Jiao Tong University

{qijunluo,mengqili1}@link.cuhk.edu.cn, l.zhao@sjtu.edu.cn,
lixiao@cuhk.edu.cn

## Abstract

Training language models on long sequence data is a demanding requirement for enhancing the model's capability on complex tasks, e.g., long-chain reasoning. However, as the sequence length scales up, the memory cost for storing activation values becomes huge during the Backpropagation (BP) process, even with the application of gradient checkpointing technique. To tackle this challenge, we propose a *memory-efficient* and *exact* BP method called **StreamBP**, which performs a linear decomposition of the chain rule along the sequence dimension in a layer-wise manner, significantly reducing the memory cost of activation values and logits. The proposed method is applicable to common objectives such as SFT, GRPO, and DPO. From an implementation perspective, StreamBP achieves less computational FLOPs and faster BP speed by leveraging the causal structure of the language model. Compared to gradient checkpointing, StreamBP scales up the maximum sequence length of BP by $2.8 - 5.5\times$ larger, while using comparable or even less BP time. Note that StreamBP's sequence length scaling ability can be directly transferred to batch size scaling for accelerating training. We further develop a communication-efficient distributed StreamBP to effectively support multi-GPU training and broaden its applicability. Our code can be easily integrated into the training pipeline of any transformer models and is available at https://github.com/Ledzy/StreamBP.

## 1 Introduction

Large language models (LLMs) [1, 6, 32] have been regarded as a powerful approach toward general artificial intelligence [3]. Recently, there has been growing interest in training LLMs for reasoning tasks, leading to significant improvements on math and code benchmarks as shown by OpenAI-o1 and DeepSeek-R1 [12, 8]. However, such models often require extremely long input sequences due to the inclusion of detailed reasoning traces [8, 35, 24, 33, 34], which results in substantial GPU memory consumption during backpropagation (BP) when training reasoning models using either reinforcement learning (RL) or supervised finetuning (SFT). This considerable memory usage mainly stems from storing intermediate activations during the BP process. This work aims to provide an efficient solution to such a memory issue.

**Background and related works.** Due to rapid growth of research in this field, we provide a non-exhaustive background overview on reasoning models and memory-efficient training methods.

---

[*]Corresponding Author

*Training LLM on long reasoning traces.* To incentivize long chain of thought (CoT) [29] reasoning capability, RL is one of most popular approaches [12, 8]. We also refer to, e.g., [7, 25, 22, 35, 19, 37, 10, 28], for replicating the "aha moment" of reasoning described in DeepSeek-R1. The core step of RL for incentivizing reasoning ability is to use a rule-based reward, e.g., the correct answer of the math question, to provide training signal towards fitting the correctly generated answer by the model itself with reasoning traces. It has been observed that as the RL training progresses, the sequence length will be increased [8]. Apart from RL techniques, it has been shown that the simple SFT training pipeline with reasoning data distilled from strong reasoning models such as R1 can also incentivize reasoning ability of LLMs; see, e.g., [24, 33, 15, 30]. The length of SFT reasoning traces varies widely, ranging from 8k to 32k, and in some cases exceeding 100k. Training models on these long traces requires significant memory cost in storing activation values. The essential roles of RL and SFT for incentivizing reasoning ability are still under active discussion [36].

*Memory-efficient training methods.* Existing memory-efficient solution for training LLMs mainly focus on designing parameter-efficient or low-dimensional version of Adam [13, 20], significantly reducing the memory usage of storing the gradient and optimizer states. Typical algorithms include LoRA [9] and its variants, e.g., [5, 18], GaLore [38], BAdam [23], etc. However, in the context of training reasoning models with long reasoning sequences, the dominant source of memory consumption arises from the BP process for storing intermediate activations. Gradient checkpointing technique [4] reduces the memory cost by recomputing the activation during the backward process. However, it still stores the full activation of the reforwarded layer and the full logits of the output layer; see Figure 1 for an illustration. MsT [21] reduces the memory cost of SFT logits by iteratively processing mini-sequence. However, it still requires the full storage of layer activations during reforward and does not apply to RL objectives. BPT [17] reduces the activation memory in transformer layer by partitioning the forward process across the sequence dimension, yet it does not handle the logits memory cost in language modeling head layer. Sequence parallel approaches increase the maximum sequence length by partitioning sequence across GPUs [16, 11, 14], which requires access of multiple GPUs. MoNET [26] explores operator-level memory optimization, yet it is specifically designed for vision models and cannot be applied to the training of LLM.

**Main contributions.** Motivated by the above observations, this work aims to address the memory issue occurred during the BP process. Our main contributions are summarized below.

(C.1) We propose a *memory-efficient* and *exact* backpropagation algorithm called **StreamBP**, which significantly reduces memory usage of storing the intermediate activation values when training LLMs on ultra-long sequences, e.g., training reasoning models. StreamBP is based on a linear decomposition of the chain rule and involves nontrivial and intricate developments for the language modeling head and transformer layers. As a result, StreamBP is compatible with common training objectives, including SFT, GRPO, and DPO. Moreover, by leveraging the causal structure of LLM, StreamBP is able to save computational FLOPs compared to the standard BP, achieving faster BP speed than the gradient checkpointing baseline. To support multi-GPU training, we also develop a distributed implementation of StreamBP with special attention to gradient and parameter communication, significantly improving training efficiency and broadening its applicability.

(C.2) Empirically, we measure the maximum sequence length and time cost of StreamBP under both single GPU and distribuetd training settings. Specifically, we measure the memory and time cost of BP in Section 4.1. StreamBP substantially scales up the maximum sequence length by $2.8 - 5.5\times$ compared to the gradient checkpointing baseline across different model scales while achieving comparable or even less BP time. In Section 4.2, we verify the effectiveness of StreamBP under various training objectives, showing that it significantly increases maximum sequence length under the objective of SFT, GRPO, and DPO. In Section 4.3, we show that under Deepspeed distributed training scheme, StreamBP is able to achieve about $5 - 5.6\times$ larger sequence length than distributed gradient checkpointing.

## 2 Preliminary: Peak Memory Cost during Backpropagation

As a concrete example, we utilize PyTorch CUDA memory snapshot tool to record detailed GPU memory usage across 2 forward and backward processes of Qwen 3-4B model under the sequence length of 8192, where the gradient checkpointing technique is applied. The result is shown in Figure 1. We exclude the memory cost of optimizer states and focus solely on the BP process.

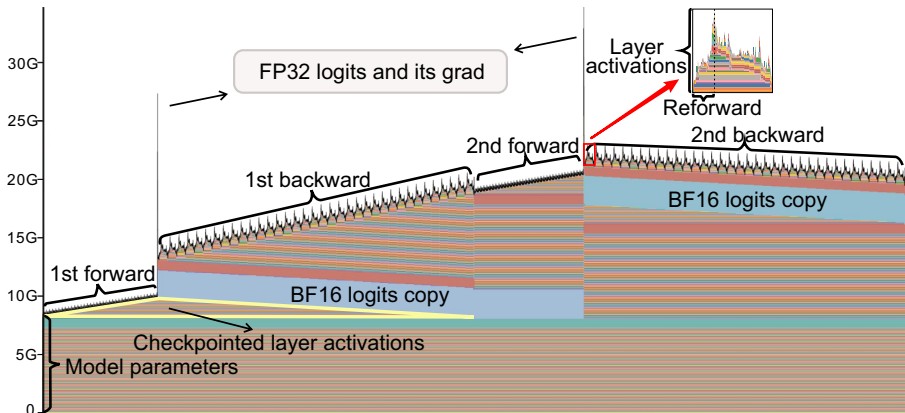

Figure 1: Memory profile during backpropagation of Qwen 3-4B with gradient checkpointing, visualized using PyTorch's memory profiler. Sequence length is set to 8192. Batch size is 1. Optimizer states are excluded as we focus on the BP process.

**Peak memory cost.** The peak memory cost occurs at the end of the 2nd forward pass. Apart from the 16GB allocated for BF16 parameters and gradients, approximately 14GB is used for storing intermediate computations. This includes BF16 checkpointed layer inputs, BF16 logits, FP32 logits, and the gradient of FP32 logits. Since the logits is of dimension sequence length ($T$) × vocabulary size ($C$), their memory cost is independent of the model size and is subject to $C$ of the model class.

**Second peak memory cost.** The 2nd memory peak happens at the start of the 2nd backward, where the model reforward the checkpointed inputs of the last transformer layer to compute layer activations. The activation values will be temporarily stored for computing the gradient of the layer's parameter and inputs.

We remark that as the model size grows up, the 2nd peak memory will increase and become closer to the peak memory. We provide more detailed explanation of the memory profile in Appendix C.

## 3 Memory-Efficient Exact Stream Backpropagation

In this section, we introduce the design of the proposed algorithm, which computes *exact* gradient with reduced memory cost and floating point operations (FLOPs) during the BP process.

### 3.1 Main Idea

Consider a transformation happened during the model's forward pass $f_W(Z_{\text{in}}) = Z_{\text{out}}$, where $W$ is the weight associated with the transformation, $f_W(\cdot)$ can be any mapping inside the model, e.g., a transformer layer or the language modeling head. By chain rule, the gradient of weight are given by $\frac{\partial L}{\partial \text{vec}(W)} = \left(\frac{\partial \text{vec}(Z_{\text{out}})}{\partial \text{vec}(W)}\right)^\top \frac{\partial L}{\partial \text{vec}(Z_{\text{out}})}$, where $L$ is the loss, $\text{vec}(\cdot)$ is the vectorized operator, and $\partial \text{vec}(Z_{\text{out}})/\partial \text{vec}(W)$ denotes the Jacobian matrix. During BP, when the gradient $\partial L/\partial \text{vec}(Z_{\text{out}})$ is ready, gradient checkpointing method will reforward $Z_{\text{in}}$ through $f_W(\cdot)$, and then calculate and store all the intermediate activations that are required for computing $\partial \text{vec}(Z_{\text{out}})/\partial \text{vec}(W)$ and $\partial \text{vec}(Z_{\text{out}})/\partial \text{vec}(Z_{\text{in}})$, where $\partial \text{vec}(Z_{\text{out}})/\partial \text{vec}(Z_{\text{in}})$ will be used for calculating $\partial L/\partial \text{vec}(Z_{\text{out}})$ for the preceding layer. As shown in Figure 1, the memory cost of storing these intermediate activation values can be huge.

To reduce the memory cost of these intermediate values, we introduce the **stream backpropagation (StreamBP)**. Let $\text{vec}(Z_{\text{out}}) = [\text{vec}(Z_{\text{out}}^{(1)}), \text{vec}(Z_{\text{out}}^{(2)}), \ldots, \text{vec}(Z_{\text{out}}^{(D)})]$ be any partition of $\text{vec}(Z_{\text{out}})$. StreamBP is based on the following linear decomposition:

$$\frac{\partial L}{\partial \text{vec}(W)} = \left(\frac{\partial \text{vec}(Z_{\text{out}})}{\partial \text{vec}(W)}\right)^\top \frac{\partial L}{\partial \text{vec}(Z_{\text{out}})} = \sum_{i=1}^{D} \left(\frac{\partial \text{vec}(Z_{\text{out}}^{(i)})}{\partial \text{vec}(W)}\right)^\top \frac{\partial L}{\partial \text{vec}(Z_{\text{out}}^{(i)})}. \tag{1}$$

By strategically partitioning $\text{vec}(Z_{\text{out}})$, storing the intermediate activations required for computing $\partial\text{vec}(Z_{\text{out}}^{(i)})/\partial\text{vec}(W)$ can be significantly cheaper than storing those required for calculating $\partial\text{vec}(Z_{\text{out}})/\partial\text{vec}(W)$, and is often proportional to the partitioned chunk size. Motivated by this fact, StreamBP sequentially computes the decomposed components in (1) across all the partitions, and accumulates them in a running sum, yielding the exact gradient. We refer to Appendix A.1 for a quick grasp of StreamBP's idea when using linear transformation.

Next, we elaborate on applying StreamBP to a concrete transformer LLM, which necessitates nontrivial and intricate developments.

## 3.2 StreamBP for Transformer LLMs

In this section, we apply StreamBP to significantly reduce the memory cost consumed by language modeling head and transformer layer during BP. Additionally, we will discuss that StreamBP requires even less computational FLOPs compared to the standard BP with gradient checkpointing, enabling potential time acceleration over standard approaches.

### 3.2.1 StreamBP for Language Modeling Head: SFT, GRPO, and DPO

The language modeling head performs the following linear transformation:

$$HW_{\text{lm\_head}} = \text{logits}.$$

Here, $W_{\text{lm\_head}} \in \mathbb{R}^{d \times C}$ is the weight of language modeling head, $H \in \mathbb{R}^{T \times d}$ is the hidden states output of the last transformer layer. The logits $\in \mathbb{R}^{T \times C}$ will be used to compute the objective function. $C, d, T$ are the vocabulary size, hidden dimension, and sequence length, respectively. As shown in Figure 1, the logits and its gradient give rise to a huge memory consumption, due to the large vocabulary size and sequence length. In the following, we analyze one-by-one how the memory of logits can be significantly reduced using StreamBP in the regime of SFT, GRPO, and DPO.

**Supervised finetuning (SFT).** The (un-normalized) "code-style" objective of SFT is given by

$$L_{\text{SFT}}(\text{logits}, Y) := \sum_{t=1}^{T-1} - \log \text{softmax}(\text{logits}_{t,:})_{Y_t}, \tag{2}$$

where $Y \in \mathbb{R}^{T-1}$ is the label vector. Importantly, each position's logits contribute to the objective independently. To perform StreamBP, the logits and label are evenly partitioned into $D$ chunks across the sequence dimension, i.e., $\{(\text{logits}^{(i)}, Y^{(i)}) | i = 1, \ldots, D\}$ with $\text{logits}^{(i)} \in \mathbb{R}^{((T-1)/D) \times C}$. Then, we sequentially accumulate the gradient across all partitions $i = 1, \ldots, D$ as

$$g_{\text{lm\_head}} \mathrel{+}= \frac{\partial L_{\text{SFT}}(\text{logits}^{(i)}, Y^{(i)})}{\partial W_{\text{lm\_head}}}, \quad g_H \mathrel{+}= \frac{\partial L_{\text{SFT}}(\text{logits}^{(i)}, Y^{(i)})}{\partial H}, \tag{3}$$

where $g_{\text{lm\_head}}$ and $g_H$ are initialized from zero. The operator $\mathrel{+}=$ denotes the in-place summation. $\text{logits}^{(i)}$ and its gradient will be cleaned from memory once they have been used in (3). After the accumulation across all partitions, $g_{\text{lm\_head}}$ and $g_H$ will be the exact gradient of the $W_{\text{lm\_head}}$ and $H$, respectively. During this computation, StreamBP only stores $\text{logits}^{(i)}$ and its gradient sequentially for all $i$, which only costs $1/D$ memory compared to the original approach.

**Group relative policy optimization (GRPO).** The principle of StreamBP for GRPO aligns with SFT. For compact presentation, we display the detailed derivation in Appendix A.2.

**Direct preference optimization (DPO).** The objective of DPO is given by

$$L_{\text{DPO}}(\text{logits}) := -\mathbb{E}\left[\log \sigma\left(\beta \sum_{t=1}^{T} \underbrace{\left(\log \frac{\pi_\theta(y_{w,t}|x, y_{w,<t})}{\pi_{\text{ref}}(y_{w,t}|x, y_{w,<t})} - \log \frac{\pi_\theta(y_{l,t}|x, y_{l,<t})}{\pi_{\text{ref}}(y_{l,t}|x, y_{l,<t})}\right)}_{\triangleq \ell(t)}\right)\right]. \tag{4}$$

Here, $\text{logits} := \{\text{logits}_{\pi_\theta}, \text{logits}_{\pi_{\text{ref}}}\}$ with $\text{logits}_\pi \in \mathbb{R}^{T \times C}$, $\sigma(\cdot)$ is the sigmoid function. $T$ is the sequence length of response $y$. To ease expression, we assume $y_w$ and $y_l$ have the same length without loss of generality. The policy's output is determined by logits:

$$\pi(y_t|x, y_{<t}) = \text{softmax}(\text{logits}_{t,:})_{y_t}.$$

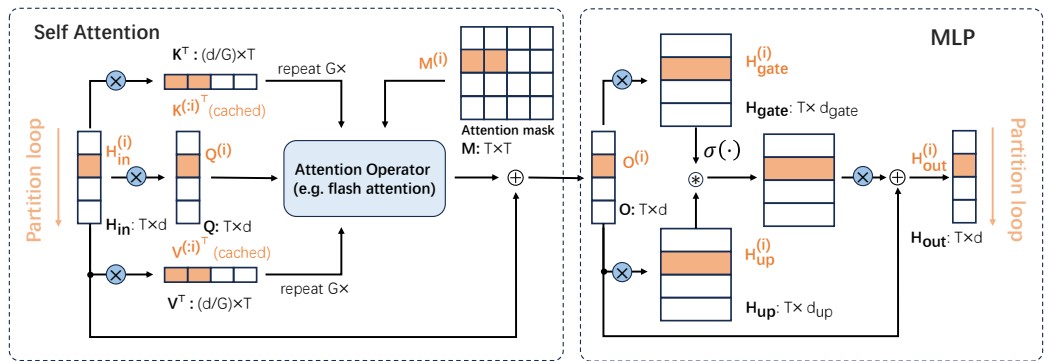

Figure 2: StreamBP for transformer layer (best view in color). The stored activations of StreamBP is highlighted in orange. Unlike the gradient checkpointing approach that reforward $H_{\text{in}}$ to compute all the activatioins required for the backward of $H_{\text{out}}$, StreamBP computes the activations only for one partition of $H_{\text{out}}^{(i)}$ at a time, which reduces the memory cost by a large margin.

Unlike SFT and GRPO, DPO's objective cannot be divided into summation of losses on individual partitions, due to the existence of the non-linear log-sigmoid transformation. Fortunately, its gradient retains a separable structure:

$$\frac{\partial L_{\text{DPO}}}{\partial W} = -\mathbb{E}_{(x,y_w,y_l)\sim\mathcal{D}}\left[\left(1-\sigma\Big(\beta\sum\nolimits_{t=1}^{T}\ell(t)\Big)\right)\beta\sum\nolimits_{t=1}^{T}\frac{\partial\ell(t)}{\partial W}\right]. \tag{5}$$

StreamBP partitions logits across the sequence dimension $\{\text{logits}^{(i)} := \{\text{logits}_{\pi_\theta}^{(i)}, \text{logits}_{\pi_{\text{ref}}}^{(i)}\}|i = 1,\ldots,D\}$ with $\text{logits}_{\pi}^{(i)} \in \mathbb{R}^{(T/D)\times C}$. Based on the partition, it performs the following accumulations for $i = 1,\ldots,D$:

$$\ell \mathrel{+}= \sum\nolimits_{t\in\mathcal{T}_i}\ell(t), \quad g_{\text{lm\_head}} \mathrel{+}= \beta\sum\nolimits_{t\in\mathcal{T}_i}\frac{\partial\ell(t)}{\partial W_{\text{lm\_head}}}, \quad g_H \mathrel{+}= \beta\sum\nolimits_{t\in\mathcal{T}_i}\frac{\partial\ell(t)}{\partial H}, \tag{6}$$

where $\mathcal{T}_i := \{(i-1)\frac{T}{D} < t \le i\frac{T}{D}|\ t \in \mathbb{Z}\}$. After finishing the above accumulation, StreamBP performs the following in-place correction to compute the exact gradient:

$$g_{\text{lm\_head}} \leftarrow (\sigma(\beta\ell)-1)g_{\text{lm\_head}}, \quad g_H \leftarrow (\sigma(\beta\ell)-1)g_H. \tag{7}$$

### 3.2.2 StreamBP for Transformer Layers: Attention and MLP

We now apply StreamBP to the transformer layer. To ease presentation, we disregard components such as normalization layer, multi-head mechanism, and residual connection without loss of generality.

A transformer layer consists of two consecutive transformations of attention and MLP:

$$O = f_{\text{attn}}(H_{\text{in}}), \quad H_{\text{out}} = f_{\text{MLP}}(O), \tag{8}$$

where $H_{\text{in}} \in \mathbb{R}^{T\times d}$ and $H_{\text{out}} \in \mathbb{R}^{T\times d}$ are the input and output of the transformer layer, respectively. The two transformations are given by

$$\textbf{Attention:} \quad Q = H_{\text{in}}W_q, \quad K = H_{\text{in}}W_k, \quad V = H_{\text{in}}W_v, \quad Q,K,V \in \mathbb{R}^{T\times d}$$
$$S = QK^\top \in \mathbb{R}^{T\times T}, \quad P = \text{softmax}(S,M) \in \mathbb{R}^{T\times T}, \quad O = PV \in \mathbb{R}^{T\times d}.$$
$$\textbf{MLP:} \quad H_{\text{up}} = OW_{\text{up}} \in \mathbb{R}^{T\times d_{\text{up}}}, \quad H_{\text{gate}} = OW_{\text{gate}} \in \mathbb{R}^{T\times d_{\text{up}}}$$
$$H_{\text{out}} = (\sigma(H_{\text{gate}}) \circ H_{\text{up}}) W_{\text{down}} \in \mathbb{R}^{T\times d}. \tag{9}$$

Computing $\partial H_{\text{out}}/\partial H_{\text{in}}$ requires storing $Q, K, V, M, O, H_{\text{up}}, H_{\text{gate}}, H_{\text{out}}$, where $M \in \mathbb{R}^{T\times T}$ is the attention mask, which is required when using mini-batch training or techniques such as sliding window attention. The storage of $S$ and $P$ can be avoided using flash attention's tiling approach.

Define the partition of quantity $H \in \mathbb{R}^{T\times d}$ along the sequence dimension as $\{H^{(i)}|i = 1,\ldots,D\}$ with $H^{(i)} \in \mathbb{R}^{(T/D)\times d}$. Let $H^{(:i)} \in \mathbb{R}^{(iT/D)\times d}$ be the concatenation of $\{H^{(j)}\}_{j=1}^{i}$ along the sequence dimension. StreamBP for transformer layer is built upon the following observation:

**Property 3.1** *The computation of $\partial H_{\text{out}}^{(i)}/\partial W$ only depends on $O^{(i)}$, $Q^{(i)}$, $K^{(:i)}$, and $V^{(:i)}$.*

To justify the above property, StreamBP sequentially performs the following partitioned attention and MLP for each chunk $i$:

**Partitioned Attention:** $\quad Q^{(i)} = H_{\text{in}}^{(i)} W_q, \quad K^{(:i)} = H_{\text{in}}^{(:i)} W_k, \quad V^{(:i)} = H_{\text{in}}^{(:i)} W_v$

$$S^{(i)} = Q^{(i)} K^{(:i)^\top}, \quad P^{(i)} = \text{softmax}(S^{(i)}, M^{(i)}), \quad O^{(i)} = P^{(i)} V^{(:i)}$$

**Partitioned MLP:** $\quad H_{\text{up}}^{(i)} = O^{(i)} W_{\text{up}}, \quad H_{\text{gate}}^{(i)} = O^{(i)} W_{\text{gate}}$

$$H_{\text{out}}^{(i)} = \left( \sigma(H_{\text{gate}}^{(i)}) \circ H_{\text{up}}^{(i)} \right) W_{\text{down}} \in \mathbb{R}^{(T/D) \times d}. \tag{10}$$

When the partitioned reforward finished, the activation for computing $\partial \text{vec}(H_{\text{out}}^{(i)})/\partial \text{vec}(W)$ is stored in memory. Then, we accumulate the gradient for $i = 1, \ldots, D$ as

$$\text{vec}(g_W) \mathrel{+}= \frac{\partial \text{vec}(H_{\text{out}}^{(i)})}{\partial \text{vec}(W)}^\top \frac{\partial L}{\partial \text{vec}(H_{\text{out}}^{(i)})}, \quad \text{vec}(g_{H_{\text{in}}}) \mathrel{+}= \frac{\partial \text{vec}(H_{\text{out}}^{(i)})}{\partial \text{vec}(H_{\text{in}})}^\top \frac{\partial L}{\partial \text{vec}(H_{\text{out}}^{(i)})}. \tag{11}$$

When the accumulation is finished, $g_W$ and $g_{H_{\text{in}}}$ become the exact gradient of weight matrices $W := \{W_q, W_k, W_v, W_{\text{up}}, W_{\text{gate}}, W_{\text{down}}\}$ and $H_{\text{in}}$, respectively. Note that $K^{(:i)}$ and $V^{(:i)}$ are needed for each partitioned forward. Hence, we compute $K$ and $V$ at once and cache it during StreamBP of the current layer. We remark that the partitioned attention is compatible with the flash attention. We illustrate the StreamBP for transformer layer in Figure 2.

**Memory efficiency of StreamBP.** StreamBP only needs to store the following activation values: $Q^{(i)}$, $K$, $V$, $M^{(i)}$, $O^{(i)}$, $H_{\text{up}}^{(i)}$, $H_{\text{gate}}^{(i)}$, and $H_{\text{out}}^{(i)}$. Note that when grouped query attention [2] is used with group size $G$, $K$ and $V$ only costs $1/G$ memory of $Q$. Consequently, StreamBP costs approximately $1/D$ memory for activation value compared to the standard BP.

**Computational efficiency of StreamBP.** For long sequence training, the most computational expensive operation involved in transformer layer is the calculation of the pre-attention score $S$. StreamBP reduces the FLOPs of the operation by approximately half. Specifically, the standard implementation $S = QK^\top$ costs FLOPs of $2T^2 d^2$, while StreamBP only costs FLOPs of $\frac{(1+D)T^2 d^2}{D}$ for computing $S^{(i)} = Q^{(i)} K^{(:i)^\top}$ across $D$ partitions. The FLOPs reduction is since that StreamBP utilizes the causal structure of language models, which uses $K^{(:i)}$ rather than $K$ in the computation of $S^{(i)}$. For all the other operations, StreamBP across all partitions shares the same FLOPs as the standard BP with checkpointing. Note that $K$ and $V$ are only computed once and get cached.

**HBM overhead of StreamBP.** Each partitioned gradient calculation in (1) requires loading model weight $W$ from high bandwidth memory (HBM) to register for computation, which induces additional overhead compared to the standard BP. Meanwhile, StreamBP reduces the HBM throughput of attention mask by approximately half. The overhead directly depends on the number of partitions $D$. In Section 4.5, we empirically study how $D$ affects the BP time and memory cost.

### 3.3 Distributed StreamBP

Although StreamBP is naturally compatible with modern distributed training techniques such as distributed data parallel (DDP) and Deepspeed ZeRO, directly applying these techniques to StreamBP is inefficient, due to redundant communications of gradients and parameters. We display the design of distributed StreamBP in Figure 6. The major designs contain gradient communication design and parameter communication design.

**Gradient communication.** During the backward, when the local gradient buffer of all processes are ready, a gradient averaging operation (i.e. all-reduce or reduce-scatter) is performed across all the processes. As shown in (1), a parameter's gradient in StreamBP is ready until the accumulation operation is finished. To avoid redundant gradient communication, the gradient averaging of distributed StreamBP is performed after the accumulation. In this sense, distributed StreamBP will have the same gradient communication cost as the standard BP.

**Parameter communication.** Parameter communication is required when using model-parallel. For instance, ZeRO-3 partitions each parameter evenly across different GPUs, and gather the parameter

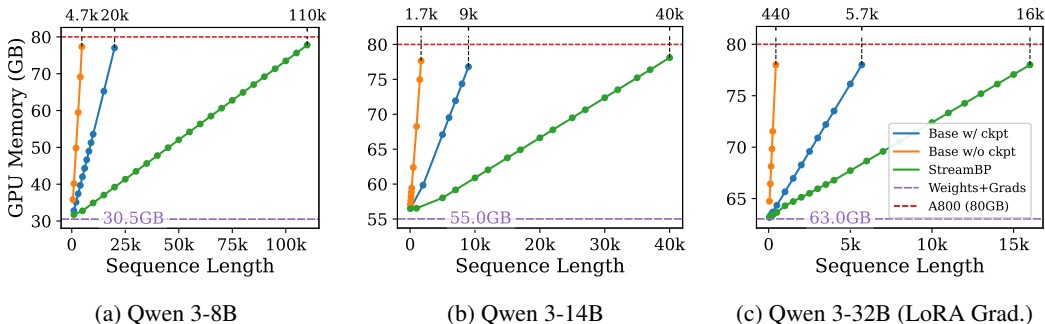

Figure 3: Peak BP memory cost measurement of Qwen 3-8B, 14B, and 32B models under different sequence lengths. Under the 80GB memory limit, StreamBP scales the maximum sequence by $2.8 - 5.5\times$ larger compared to gradient checkpointing.

| Sequence length | 6k | 9k | 12k | 15k | 18k | 21k | 24k | 27k |
|---|---|---|---|---|---|---|---|---|
| Baseline w/o ckpt | 2.0 | – | – | – | – | – | – | – |
| Baseline w/ ckpt | 2.5 | 4.7 | 7.4 | 10.8 | 14.7 | 19.3 | 24.3 | – |
| MsT | 2.6 | 4.9 | 7.6 | 11.2 | 15.2 | 20.3 | 25.2 | 31.8 |
| **StreamBP** | 2.6 | 4.7 | 7.0 | 10.0 | 13.2 | 17.2 | 21.2 | 25.8 |
| Acceleration over ckpt | –4.4% | 0.4% | 6.0% | 7.4% | 10.5% | 10.9% | 12.9% | – |

Table 1: BP Time cost (in seconds) under different sequence lengths. The result is based on Qwen 3-4B model and is averaged over 50 independent trials. The acceleration of StreamBP becomes more apparent as the sequence length grows up, corroborating our analysis in Section 3.2.2.

when the operators associated with it is called. By (1), StreamBP requires the access of unpartitioned parameter for $D$ times during the backward. Hence, naively implementing model-parallel for StreamBP costs $(D - 1)\times$ all-gather operations. To reduce the communication cost, distributed StreamBP caches the weight locally until the accumulation of its gradient is finished, avoiding inducing additional parameter communication cost compared to the standard BP. The design will introduce an additional memory cost of storing a single transformer layer in each GPU.

## 4 Experiments

In this section, we evaluate StreamBP based on the Qwen 3 model series. The result directly applies for any causal language models such as Llama, Mistral, and Gemma, since StreamBP is not restricted to a certain model class. Our evaluation mainly consists of 3 parts, including 1) **backpropogation cost**; 2) **training cost**; and 3) **distributed training**. All the experiments are conducted using A800-80GB GPUs. The detailed setup is presented in Appendix D.

### 4.1 Backpropagation Cost Measure

**Memory cost.** We report the memory cost of StreamBP and baseline (i.e., standard BP) with/without gradient checkpointing in Figure 3. Given the 80GB memory limit, StreamBP is able to increase the maximum sequence length by 2.8-5.5× and 23.4-36.3× compared to baseline with/without gradient checkpointing, respectively. Importantly, the memory cost of StreamBP is linearly related to the sequence length, as StreamBP does not store the full attention mask and is compatible with flash attention. Thus, StreamBP's sequence length scaling can be directly transferred to batch size scaling for accelerating the training. Note that the ratio differs across different model sizes, which attributes to distinct configurations of hidden dimensions.

**Time cost.** We compare the BP time cost of StreamBP with baseline approaches, and present the result in Table 1. The result shows that StreamBP consistently exhibits faster BP time compared to the gradient checkpointing baseline across a wide range of sequence lengths, and beats the long-sequence training baseline MsT [21] more significantly. As the sequence length increases, the acceleration

| Objective | Method | 4B | | 8B | | 14B | 32B |
|---|---|---|---|---|---|---|---|
| | | Full | LoRA | Full | LoRA | LoRA | LoRA |
| SFT | Baseline w/o ckpt | 7.0 | 7.1 | 3.4 | 5.1 | 2.9 | 0.4 |
| | Baseline w/ ckpt | 28.5 | 36.5 | 15.7 | 30.6 | 23.0 | 5.9 |
| | **StreamBP** | 200.0 | 247.0 | 72.0 | 142.4 | 84.6 | 16.3 |
| GRPO | Baseline w/o ckpt | 0.9 | 1.0 | – | 0.7 | 0.4 | – |
| | Baseline w/ ckpt | 8.5 | 12.1 | – | 9.5 | 6.9 | – |
| | **StreamBP** | 18.2 | 27.5 | – | 16.5 | 10.1 | – |
| DPO | Baseline w/o ckpt | 3.1 | 3.5 | – | 2.5 | 1.4 | 0.2 |
| | Baseline w/ ckpt | 18.7 | 32.2 | – | 25.5 | 18.5 | 4.4 |
| | **StreamBP** | 57.2 | 100.2 | – | 60.9 | 36.9 | 7.8 |

Table 3: Maximum sequence length (in thousands) on a single A800-80GB GPU.

becomes more significant. The result aligns with our analysis in Section 3.2.2, where StreamBP reduces approximately half of computation in calculating attention scores, whose computational FLOPs is quadratically related to the sequence length. In Table 2, we show that StreamBP's memory-efficiency enables the usage of much larger batch size, which helps further accelerate the BP speed. The acceleration attributes to the reduced sample-wise HBM throughput of model loading and better leveraging of parallel processing. In particular, StreamBP with batch size 16 achieves even less per-sample BP time than the baseline without gradient checkpointing under batch size 1.

| Batch Size | 1 | 2 | 4 | 16 |
|---|---|---|---|---|
| Baseline w/ ckpt | 7.42 | 7.12 | 7.04 | – |
| MsT | 7.64 | 7.41 | 7.28 | 7.06 |
| **StreamBP** | 7.03 | 6.82 | 6.63 | 6.47 |

Table 2: Per-sample BP time cost of Qwen 3-4B model under different batch sizes. The sequence length is 9000. StreamBP achieves further acceleration by utilizing substantially larger batch size.

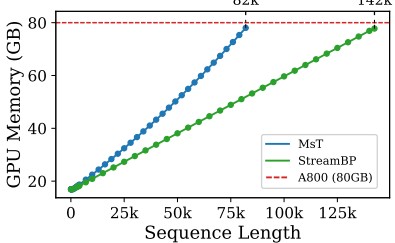

Figure 4: Memory cost comparison with MsT.

## 4.2 Training Cost Measure

**Sequence length scaling.** We present the maximum sequence length of StreamBP and baseline methods given a single A800-80GB GPU in Table 3. For all the objectives, StreamBP significantly increases the maximum sequence length over baseline approaches, which justifies the efficiency of StreamBP in reducing the memory cost of transformer layer activations and logits. For SFT, even 32B model can achieve sequence length up to 16k. Note that the sequence length scaling ratio can be equally transferred to batch size scaling ratio for acceleration. For example, for the SFT of 8B model, StreamBP enables $\frac{72}{15.7} \approx 4.5\times$ larger batch size than the baseline with gradient checkpointing.

**Comparison with long-sequence training baseline.** We compare StreamBP with the long-sequence SFT baseline MsT, and plot the peak memory of training Qwen 3-8B LoRA model in Figure 4. The result shows that StreamBP achieves approximately $1.7\times$ larger sequence length than MsT. In terms of time-efficiency, StreamBP requires significantly less BP time compared to MsT, as shown in Table 1. The acceleration becomes more significant as the sequence length scales up.

## 4.3 Efficiency under Distributed Training

We measure the maximum sequence scaling and BP time under the Deepspeed ZeRO-2 SFT scheme, where the gradient and optimizer states are partitioned across different GPUs and each GPU processes its local data batch. The results are shown in Table 4 and Table 5. Compared to baseline with gradient checkpointing, distributed StreamBP scales to a maximum sequence length approximately 5-5.6× larger and achieves noticeably faster BP speed.

| # of GPUs | 3 | 4 | 5 | 6 | 7 | 8 |
|---|---|---|---|---|---|---|
| Baseline w/ checkpoint | 18.6 | 20.5 | 22.3 | 23.1 | 23.5 | 23.7 |
| **Distributed StreamBP** | 92.4 | 112.0 | 121.5 | 128.7 | 131.4 | 131.8 |

Table 4: Maximum sequence length (in thousands) of Qwen 3-8B under ZeRO-2 training scheme.

| Sequence length | 6k | 9k | 12k | 15k | 18k | 21k |
|---|---|---|---|---|---|---|
| Baseline w/ checkpoint | 8.4 | 9.6 | 12.2 | 18.5 | 22.9 | 26.3 |
| **Distributed StreamBP** | 6.4 | 8.3 | 10.8 | 14.8 | 17.2 | 20.7 |

Table 5: Per-sample BP time cost (in seconds) of Qwen 3-8B under ZeRO-2 training scheme.

### 4.4 Correctness Verification of StreamBP

We empirically measure the gradient difference of StreamBP and baseline methods to justify the correctness of our implementation. Importantly, the float point operations has the following associativity issue due to numerical precision:

$$(a \oplus b) \oplus c \neq a \oplus (b \oplus c) \tag{12}$$

where $\oplus$ denotes floating-point addition. Therefore, it is impossible for the gradient difference to be zero given the different computation order. To verify the correctness, we use the gradient computed by the pure FP32 BP of the standard BP with gradient checkpointing as ground truth, denoted as `base32`. Then, we calculate its difference with the gradient $g_i$ obtained by pure BF16 BP of StreamBP and baseline BP, respectively. Additionally, we include a StreamBP run under FP32 precision, denoted as `stream32`, to verify its numerical equivalence to `base32`. Specifically, we define the absolute error and relative error as

$$\text{Er}_{\text{abs}}(g) = \frac{1}{n} \sum_i |g_i^{\text{base32}} - g_i|, \quad \text{Er}_{\text{rel}}(g) = \frac{1}{n} \sum_i \frac{|g_i^{\text{base32}} - g_i|}{|g_i^{\text{base32}} + 10^{-10}|}. \tag{13}$$

Table 6 reveals two key findings: 1) `stream32`'s micro-scale deviations ($\text{Er}_{\text{abs}} \sim 10^{-9}$) validate mathematical equivalence to `base32`, and 2) `stream16` demonstrates $\leq 0.04\%$ relative deviation from `base16`, indicating that StreamBP exhibits no precision loss compared to standard `bfloat16` computation.

We further verify the correctness of StreamBP via model training and display the loss of SFT, DPO, and GRPO in Table 7. The loss of StreamBP closely align with the standard BP. The tiny difference is due to different order of float operations discussed above.

### 4.5 Ablation Study and Additional Experiments

**Effect of partition size.** We fix the partition size of transformer layer to be 0.5k, 1k, 2k, and 5k, examining the memory and time cost across wide range of sequence lengths. The result is shown in Figure 5. Note that partition will not be performed if it is larger than the sequence length. When the sequence length is relatively small, e.g., less than 10k, the BP times under different partition sizes are close. However, as the sequence length scales up, using a partition size that is too small introduces substantial overhead. The overhead attributes to the additional HBM throughput on loading model weight from HBM to register for computation and repetitive kernel launches. Fortunately, as shown in Figure 5b, large partition size only introduces marginal additional memory cost. Hence, for long sequence training, one can use a relatively large partition size to maximize training efficiency.

**Additional experiments.** We provide more experiments in Appendix B and Appendix C.2. Here are the summarized results: 1) We present the sequence scaling using a single RTX3090-24GB GPU. The result shows that StreamBP is able to scale up the maximum sequence length to 15k, which is about $4.4\times$ larger compared to gradient checkpointing. 2) We present the memory profile of StreamBP in Figure 8. The profile demonstrates a substantial memory reduction in logits and activation values during layer reforwarding, which is consistent with our analysis in Section 3. 3)

|  | Er$_{abs}$ | | | Er$_{rel}$ | | |
|---|---|---|---|---|---|---|
| **Module** | base16 | stream16 | stream32 | base16 | stream16 | stream32 |
| lm_head | 2.56e-7 | 2.64e-7 | 6.66e-9 | 1.43% | 1.47% | 0.04% |
| Transformer layers | 2.46e-5 | 2.47e-5 | 1.75e-7 | 6.63% | 6.59% | 0.04% |

Table 6: Gradient precision analysis. StreamBP is the numerically correct and has nearly identical gradient as the standard BP.

| | SFT (Capybara) | | | DPO (Ultrafeedback) | | | GRPO (TL;DR) | | |
|---|---|---|---|---|---|---|---|---|---|
| **Step** | **Base** | **StreamBP** | **Diff.** | **Base** | **StreamBP** | **Diff.** | **Base** | **StreamBP** | **Diff.** |
| 1 | 5.782 | 5.782 | +0.000 | 1.702 | 1.702 | +0.000 | 0.013 | 0.013 | +0.0000 |
| 100 | 4.924 | 4.926 | +0.002 | 1.087 | 1.090 | +0.002 | 0.011 | 0.011 | -0.0002 |
| 200 | 4.313 | 4.309 | -0.003 | 0.938 | 0.935 | -0.003 | 0.005 | 0.004 | -0.0004 |
| 300 | 3.990 | 3.993 | +0.003 | 0.876 | 0.879 | +0.003 | 0.006 | 0.006 | +0.0000 |
| 400 | 3.791 | 3.795 | +0.003 | 0.836 | 0.840 | +0.004 | 0.005 | 0.005 | +0.0001 |
| 500 | 3.667 | 3.663 | -0.004 | 0.801 | 0.796 | -0.004 | 0.004 | 0.004 | -0.0001 |

Table 7: Evaluation loss of StreamBP and standard BP under different objectives.

We conduct experiments on Llama and Gemma architectures in Appendix B.2, demonstrating that StreamBP is applicable to general transformer models.

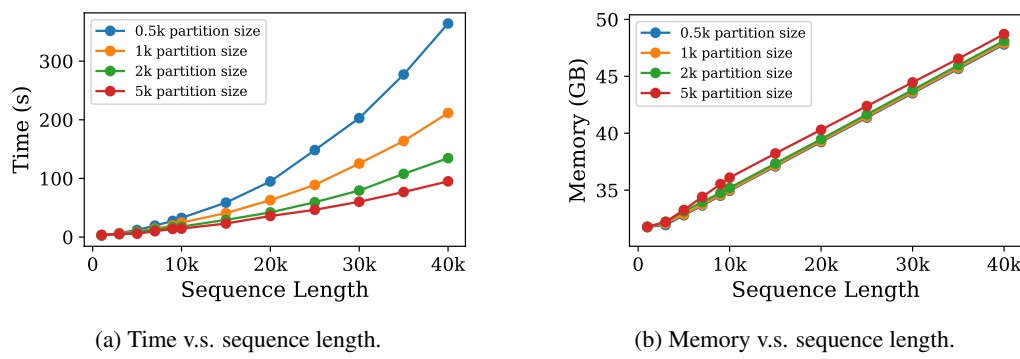

(a) Time v.s. sequence length.
(b) Memory v.s. sequence length.

Figure 5: Time and memory costs of StreamBP on Qwen 3-8B with varying partition sizes.

## 5 Conclusion and Discussions on Limitations

In this work, we developed a memory-efficient and exact BP method called StreamBP. Compared to gradient checkpointing baseline, StreamBP requires significantly less memory cost and enjoys faster BP time by leveraging the causal structure of LLMs. StreamBP can be used for long sequence training of any transformer LLMs, which finds wide applications such as training reasoning models. We also developed a communication-efficient distributed StreamBP to support multi-GPU training.

**Limitations.** Currently, StreamBP does not support MoE or multimodal models. Nonetheless, these can be addressed with simple implementation extensions, as the underlying principle remains unchanged. Additionally, StreamBP's partition size can have a clear impact on BP time. This overhead can be mitigated by employing a fused backward operator, which significantly reduces the HBM throughput. We leave these directions for future work.

**Broader impacts.** We provide a memory-efficient and exact BP method for long sequence training of LLMs. This is a technical contribution that does not yield explicit negative societal impacts.

## Acknowledgments and Disclosure of Funding

We thank the Area Chair and the anonymous reviewers for their insightful comments and suggestions, which have significantly improved the quality of the manuscript.

Lei Zhao is supported in part by the Major Project of the National Natural Science Foundation of China (NSFC) under grant 72293582 and in part by the National Key R&D Program of China under grant 2023YFA0915202. Xiao Li is supported in part by the National Natural Science Foundation of China (NSFC) under grants 12201534, 12571330, and 12326608, in part by the 1+1+1 CUHK-CUHK(SZ)-GDSTC Joint Collaboration Fund under grant 2025A0505000049, and in part by the Shenzhen Science and Technology Program under grant RCYX20221008093033010.

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

# Contents

# A    Additional Details of Stream Backpropagation

## A.1    StreamBP for Linear Transformation

Given the input $X = [X_1, X_2, \ldots, X_D]^\top \in \mathbb{R}^{D \times m}$ and weight matrices $W_1 \in \mathbb{R}^{m \times n}$, $W_2 \in \mathbb{R}^{n \times k}$. Consider the following two linear transformations:

$$Y = [Y_1, \ldots, Y_D]^\top := XW_1 \in \mathbb{R}^{D \times n}$$
$$Z = [Z_1, \ldots, Z_D]^\top := YW_2 \in \mathbb{R}^{D \times k}.$$

For the ease of expression, define $dM := \frac{\partial L}{\partial M}$ as the gradient of quantity $M$ with respect to the objective $L$, which can be arbitrary objective. The gradient of $Y$ and weight matrices are given by

$$dY = (dZ)W_2^\top, \quad dW_1 = X^\top(dY), \quad dW_2 = Y^\top(dZ).$$

The standard approach of calculating $dW_1$ and $dW_2$ requires storing the intermediate value $Y$ and $dY$. Based on (1), the above expression can be written as

$$dY_i = (dZ_i)W_2^\top, \quad dW_1 = \sum_{i=1}^{D} X_i^\top(dY_i), \quad dW_2 = \sum_{i=1}^{D} Y_i^\top(dZ_i).$$

Hence, one can sequentially compute $Y_i$ and $dY_i$, accumulate $X_i^\top dY_i$ and $Y_i^\top dZ_i$, then drop the $Y_i$ and $dY_i$. Compared to the standard gradient computation, this approach effectively reduces the memory cost of intermediate values to $1/D$ without introducing additional computational cost. In practice, to better utilize the parallel computation and reduce the HBM load, one can process $X$ in chunk-wise rather than sample-wise manner. Below, we demonstrate the memory efficiency of StreamBP with numerical experiment on the linear example.

**Experiment on linear example.**  We empirically measure the memory cost of StreamBP and compare it with standard BP, based on concrete linear example. Specifically, let $X \in \mathbb{R}^{10^6 \times 16384}$, $W_1, W_2 \in \mathbb{R}^{16384 \times 16384}$, $L = \sum_{j,k} Z_{j,k}$. StreamBP partitions $X$ as $\{X^{(i)} | i = 1, \cdots, D\}$ with $X^{(i)} \in \mathbb{R}^{(10^6/D) \times 16384}$. We present the memory cost and time cost of standard BP and StreamBP in Table 8.

| | Standard BP | StreamBP | | | | |
|---|---|---|---|---|---|---|
| | | $D = 20$ | $D = 50$ | $D = 100$ | $D = 200$ | $D = 500$ |
| Peak memory (GB) | 36.01 | 13.25 | 12.47 | 12.20 | 12.07 | 11.99 |
| Intermediate memory (GB) | 25.15 | 2.39 | 1.61 | 1.34 | 1.21 | 1.13 |
| Time cost (seconds) | 14.48 | 14.50 | 14.70 | 14.79 | 15.45 | 18.42 |

Table 8: Memory and time cost of standard BP and StreamBP under different partition number.

Compared to standard BP, StreamBP with $D = 20$ reduces memory cost by $63.2\%$ with almost no time overhead. As $D$ increases, the intermediate memory cost is further reduced.

## A.2    StreamBP for GRPO

The GRPO's objective is given by:

$$L_{\text{GRPO}}(\text{logits}) := \mathbb{E}_{[q \sim \mathcal{D}_q, \{o_j\}_{j=1}^{G} \sim \pi_{\text{old}}(\cdot|q)]}$$

$$\left[ \frac{1}{G} \sum_{j=1}^{G} \frac{1}{T_o} \sum_{t=1}^{T_o} \left( \min\left\{ \frac{\pi_\theta(j,t)}{\pi_{\text{old}}(j,t)} \hat{A}_{j,t}, \text{clip}\left( \frac{\pi_\theta(j,t)}{\pi_{\text{old}}(j,t)}, 1-\epsilon, 1+\epsilon \right) \hat{A}_{j,t} \right\} - \beta \log \frac{\pi_\theta(j,t)}{\pi_{\text{ref}}(j,t)} \right) \right].$$
$$\underbrace{\phantom{xxxxxxxxxxxxxxxxxxxxxxxxxxxxxxxxxxxxxxxxxxxxxxxxxxxxxxxxxxxxxxxxxxxxxxxxxxxxxxx}}_{\triangleq f(j,t)}$$
$$(14)$$

To ease presentation, in the above equation we omit the compensation term in GRPO's KL divergence term without loss of generality, as StreamBP directly applies to the KL divergence term. Here, $q$ is

the prompt sequence, $\mathcal{D}_q$ is the dataset of prompt, and $o_j$ is $j$-th response sequence with respect to $q$. $G$ and $T_o$ are the group size and the length of response. For the ease of expression, we assume all the responses are of the same length without loss of generality. $\hat{A}_{j,t}$ is the estimated advantage. $\pi_\theta$, $\pi_{\text{old}}$, and $\pi_{\text{ref}}$ are the target policy, old policy and reference policy, respectively.

Here, $\text{logits} \triangleq \{\text{logits}_{\pi_\theta}, \text{logits}_{\pi_{\text{old}}}, \text{logits}_{\pi_{\text{ref}}}\}$ contains logits generated by target policy, old policy, and reference policy, respectively, with $\text{logits}_\pi \in \mathbb{R}^{G \times T_o \times C}$. The policy's output is determined by logits, i.e.,

$$\pi(j, t) := \pi(o_{j,t}|q, o_{j,<t}) = \text{softmax}(\text{logits}_{\pi,j,t,:})_{o_{j,t}}.$$

Note that each $f(j, t)$ in (14) contributes to the objective independently and only depends on $\text{logits}_{\pi,j,t,:}$, which enables us to perform StreamBP along the sequence dimension. Specifically, let us partition the logits along the sequence dimension as $\{\text{logits}^{(i)} := \{\text{logits}_{\pi_\theta}^{(i)}, \text{logits}_{\pi_{\text{old}}}^{(i)}, \text{logits}_{\pi_{\text{ref}}}^{(i)}\}|i = 1, \ldots, D\}$ with $\text{logits}_\pi^{(i)} \in \mathbb{R}^{G \times (T_o/D) \times C}$. Define the objective of the sequence partition as

$$L_{\text{GRPO}}^{(i)}(\text{logits}^{(i)}) := -\mathbb{E}_{[q \sim \mathcal{D}_q, \{o_j\}_{j=1}^G \sim \pi_{\text{old}}(\cdot|q)]} \left[ \frac{1}{G} \sum_{j=1}^G \frac{1}{T_o} \sum_{t \in \mathcal{T}_i} f(j, t) \right], \quad (15)$$

where $\mathcal{T}_i := \{(i-1)\frac{T_o}{D} < t \leq i\frac{T_o}{D}|t \in \mathbb{Z}\}$ denotes the sequence range of the partition. We have $L_{\text{GRPO}}(\text{logits}) = \sum_{i=1}^D L_{\text{GRPO}}^{(i)}(\text{logits}^{(i)})$. Then, similar to SFT, StreamBP sequentially performs the following accumulation for $i = 1, \ldots, D$:

$$g_{\text{lm\_head}} += \frac{\partial L_{\text{GRPO}}^{(i)}(\text{logits}^{(i)})}{\partial W_{\text{lm\_head}}}, \quad g_H += \frac{\partial L_{\text{GRPO}}^{(i)}(\text{logits}^{(i)})}{\partial H}. \quad (16)$$

## A.3 Distributed StreamBP

StreamBP is naturally compatible with distributed training techniques such as Deepspeed ZeRO. However, to ensure the efficiency, the gradient communication and parameter communication needs to be customized. The design of distributed StreamBP is shown in Figure 6.

**Gradient communication design.** Gradient communication happens in almost all distributed training. During the backward, when the local gradient buffer of all processes are ready, a gradient averaging operation (i.e. all-reduce or reduce-scatter) is performed across all the processes. As shown in (1), a parameter's gradient in StreamBP is ready until the accumulation operation is finished. To avoid redundant gradient communication, the gradient averaging of distributed StreamBP is performed after the accumulation. In this sense, distributed StreamBP will have the same gradient communication cost as the standard BP.

**Parameter communication design.** Parameter communication is required when using model-parallel. For instance, ZeRO-3 partitions each parameter evenly across different GPUs, and gather the parameter when the operators associated with it is called. By (1), StreamBP requires the access of unpartitioned parameter for $D$ times during the backward. Hence, naively implementing model-parallel for StreamBP costs $(D-1)\times$ all-gather operations. To reduce the communication cost, distributed StreamBP caches the weight locally until the accumulation of its gradient is finished, avoiding inducing additional parameter communication cost compared to the standard BP. This design will introduce an additional memory cost of storing a single transformer layer in each GPU.

**Efficiency of distributed StreamBP.** The choice of batch size significantly affects the communication cost. For example, when using Deepspeed ZeRO-2, using batch size 1 with gradient accumulation step $K$ leads to approximately $K$ times heavier backward communication cost than using batch size $K$ with gradient accumulation step 1. StreamBP enables the usage of much larger batch size compared to the standard BP, which significantly reduces the communication cost. We remark ZeRO-2 reduces the above overhead by overlapping the communication and computation.

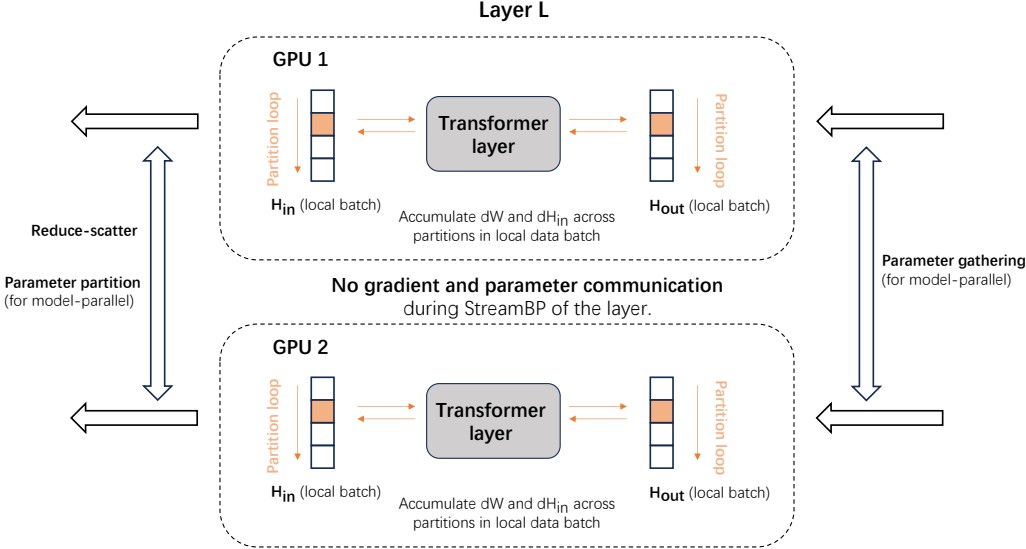

Figure 6: Design of distributed StreamBP. When backward through a transformer layer, its parameter is gathered beforehand. During the StreamBP of the layer, no gradient or parameter communication is fired. When BP of the layer is finished, the gradient will be reduced across layers and the paramater will be sharded across GPUs.

# B  Additional Experiments

## B.1  Sequence Scaling on a Single RTX3090-24GB

We present the memory cost of training Qwen 3-8B LoRA model in Figure 7. Under the 24GB memory budget, StreamBP allows sequence scaling up to 15k, which is $4.4\times$ larger than the gradient checkpointing baseline.

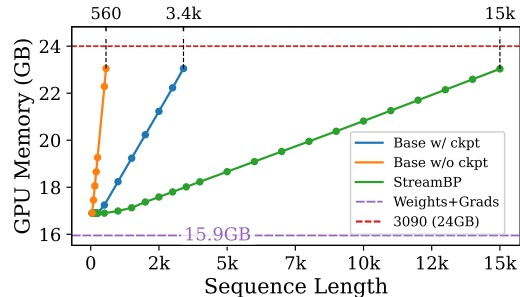

Figure 7: Peak BP memory cost measurement of Qwen 3-8B with LoRA (rank=32) on a single RTX3090-24GB GPU

## B.2  StreamBP for Broader Architectures

In this section, we display the BP memory cost of Llama 3.1-8B and Gemma 3-1B. The results are shown in Table 9.

| Model | Method | 3k | 6k | 9k | 18k |
|---|---|---|---|---|---|
| Llama 3.1-8B | Base w/o ckpt | 54.2 | – | – | – |
| | Base w/ ckpt | 35.8 | 41.7 | 47.6 | 65.6 |
| | **StreamBP** | 31.8 | 32.7 | 33.6 | 37.2 |
| Gemma 3-1B | Base w/o ckpt | 23.1 | 42.8 | 63.4 | – |
| | Base w/ ckpt | 14.3 | 24.8 | 35.3 | 67.2 |
| | **StreamBP** | 4.7 | 5.0 | 5.4 | 7.9 |

Table 9: BP Memory cost (GB) of Llama 3.1-8B and Gemma 3-1B under different sequence lengths.

Clearly, the memory cost of StreamBP is significantly less than standard BP; the memory gap increases as the sequence length scales up, which verifies the efficiency of StreamBP on different model architectures. Notably, Gemma 3-1B costs even more memory than Llama 3.1-8B for standard BP under sequence length 18k. This is because Gemma 3 model has $2\times$ larger vocabulary size than Llama 3.1, and hence doubles the memory cost of logits. In comparison, StreamBP only stores partitioned logits and its memory cost is dominated by the model weight and model gradient rather than activation values.

# C  Analysis of Memory Profile

## C.1  Memory Profile Explanation of Gradient Checkpointing

We provide detailed explanation of Figure 1. First, the "Model parameters" part stores the BF16 model parameters and constantly occupy nearly 8GB memory. Then, during the 1st forward process, gradient checkpointing gradually stores the checkpointed layer inputs, and hence it gradually consumes more memory. At the end of the 1st forward and the beginning of the 1st backward, the FP32 logits and its gradient are computed, which suddenly consumes a huge amount of memory due to the very large vocabulary size of Qwen 3. This huge memory occupation is released after calculating the gradient of lm_head (tied with embedding) as shown in the brown rectangle above the "BF16 logits

copy". During the 1st backward, BP calculates and stores the gradients of all the parameters and the memory consumption continues to increase. One interesting phenomenon is that the memory usage of the checkpointed layer activations enclosed in the yellow triangle goes to decrease during the 1st backward process. This is because that the stored corresponding activations will be deleted once the current weights' gradients are computed, as they are no longer needed. Another interesting observation is that there are some triangle-shaped bumps during the 1st backward, for which we draw a subfigure to interpret during the 2nd backward process. We explicitly show the reforward process of gradient checkpointing, and show the reforwarded layer activations that will be deleted once the corresponding gradients are computed, yielding a triangle-shaped bump.

An additional remark on Figure 1 is in order. As shown in Figure 1, the current implementation of the "Transformers" package stores an additional "BF16 logits copy" and an additional tied embedding and "lm_head" gradient in the 2nd backward process, which may be optimized as they are not used.

## C.2 Memory Profile of StreamBP

The memory profile of StreamBP is shown in Figure 8. Compared to gradient checkpointing's profile in Figure 1, StreamBP reduces the peak memory greatly by partitioning the logits across the sequence dimension. The second peak memory in reforwarded activation values is also greatly reduced, which is now only marginally higher than the storage of the key and value states.

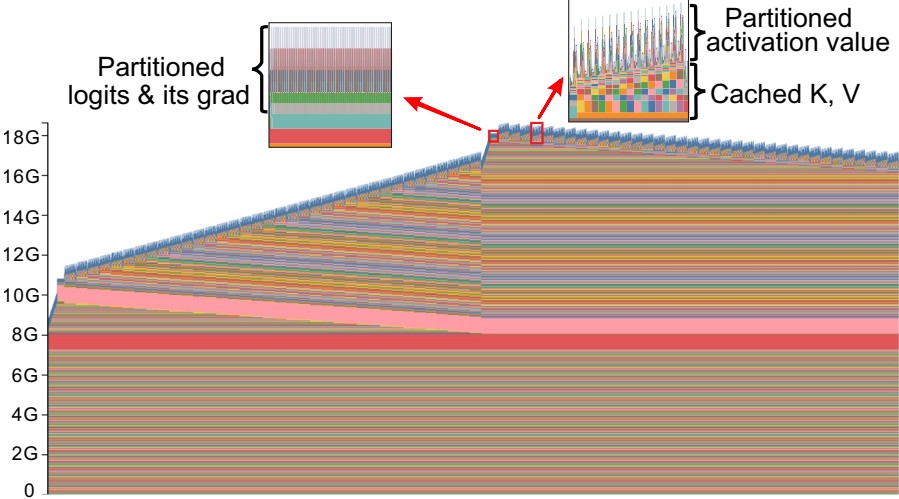

Figure 8: Memory profile with StreamBP under the same settings as Figure 1. The partition size of logits and transformer layer are set to 100 and 500, respectively.

# D   Experiment Setup

We implement our algorithm using Huggingface transformers library [31]. The detailed experiment setup is presented below.

**Backpropogation cost.** We decouple the optimization, focusing purely on the time and memory cost during the BP. This result serves as the minimum requirement on training a language model under a given sequence length. In particular, under batch size 1, we measure the BP memory cost of Qwen 3-8B, 14B and 32B models using a single A800-80GB GPU. For 32B model, we inject rank-32 LoRA adapters and only calculate the gradient of the adapters, as a single A800 GPU cannot store the model and the full gradient simultaneously. We adopt the BF16 data type for storing model weight and gradient. The partition size of language modeling head is set to 100 for all the experiments. The partition size for transformer layer is set to 500 for maximum sequence length measurement, and is set to $T/3$ for time measurement.

**Training cost.** We measure the maximum sequence length of training 4B, 8B, 14B, and 32B models using the objective of SFT, GRPO, and DPO, respectively. Our implementation is built upon HuggingFace TRL library [27]. We adopt pure BF16 Adam optimizer in full training and

mixed-precision training in rank-32 LoRA model training. When using LoRA mode, there is no need to store the reference model in DPO and GRPO, as one can disable the adapter in training model to recover the reference model. The batch size is set to 1 except for GRPO, where the group size is set to 8. We also compare the memory cost of StreamBP with long-sequence training baseline method MsT to demonstrate the effectiveness of StreamBP. The partition size of language modeling head is set to 100 for all the experiments. The partition size for transformer layer is set to 500 for maximum sequence length measurement, and is set to $T/3$ for time measurement. All the experiments are conducted in a single A800-80GB GPU.

**Distributed training.** Under the communication-efficient design in Appendix A.3, we measure the maximum sequence length and time cost of distributed StreamBP under Deepspeed ZeRO-2 training paradigm, where the gradient and optimizer states are partitioned across GPUs. Our evaluation is conducted using a single-node server with 8 A800-80GB GPUs connected by NvLink.

