# OpenReview forum: "StreamBP: Memory-Efficient Exact Backpropagation for Long Sequence Training of LLMs"
_NeurIPS.cc/2025/Conference — NeurIPS 2025 poster_

### Official Review · Reviewer_RVuF · 2025-06-01

**Clarity:** 2
**Significance:** 3
**Originality:** 2
**Rating:** 5
**Confidence:** 4

**Summary:**

The paper discusses a method to improve the memory efficiency of LLMs post-training  by partitioning the input of each layer (MLP, Attention, Output layer)  into chunks and feeding them sequentially to the layer operation. This is done in the forward pass, but also during the backward pass (using checkpointing) to save memory on the intermediate activations. For the attention operation the model precomputes the keys and values, which are reused for all chunks of the queries. The method brings significant memory and time savings when training on long sequences compared to gradient checkpointing and the competitor method MsT. This enables for example to fit the forward and backward pass of Qwen3-14B with a sequence of 48k tokens in a single 80GB GPU, while MsT and checkpointing allow only for 9k and 1.7k tokens respectively (See Figure 3). The proposed method is also faster than MsT and checkpointing thanks to a more efficient attention operation using half the flops.

**Questions:**

I am willing to increase the score depending on the answers to my questions, Q1 in particular.

**Questions and Comments**

Q1 The authors mention a reduction to half the FLOPs for their attention computation. I feel this is a core part of the approach since this is where the time savings come from, and should be discussed more and maybe put also in the contributions. I also thought that using flash-attention would already result in such a reduction in FLOPS, is that not the case? I feel it would also be useful to state clearly which attention implementation is used in the text.

Q2 The authors go into details into how the implementation of the method differs between SFT, DPO and GRPO. However, it seems that the method can be applied in the same fashion whenever the loss can be written as a sum, which is the case for GRPO and SFT. In the case of DPO instead the loss is still a nonlinear function of a scalar ($s$) which can be written as a sum, so the method is applied to $s$ and then adjusted by using the derivative of the loss w.r.t. $s$. Describing these three cases in detail seems unnecessary and the exposition could be greatly simplified by describing how the procedure can be applied whenever the function to be differentiated is a function of a scalar sum. The handling of attention is a bit more nuanced and its dedicated section could be expanded.

Q2 Lines 58-59, “nontrivial and intricate …” This part is not so clear. I would appreciate it if the authors would be more specific on this point, for example mentioning the chunking that is done by the method?

Q3 I found it interesting that the memory peaks are at the end of the second forward pass and not the first. I assume this is due to the fact that pytorch caches some memory to be reused during the forward pass. Maybe the authors can comment a bit on this point?

Minor:

M1 Figure 1: what is the batch size (I assumed one)? Maybe this could be indicated in the caption.

M2 Typo above eq. (13): I think $S^{(i)}$ should be $M^{(i)}$.

M3 Appendix A.1: $L$ is not defined.

**Ethical Concerns:**

["NO or VERY MINOR ethics concerns only"]

**Final Justification:**

The authors have adequately addressed all three points that I raised. In particular clarifying the point regarding the FLOPs saving compared to competitor methods and testing the robustness of the approach after several steps of fine-tuning. Therefore I changed the score from 4 to 5.

**Limitations:**

Yes

**Paper Formatting Concerns:**

No issues

**Quality:**

3

**Strengths And Weaknesses:**

**Strengths**

S1 The method is conceptually simple and could be easily implemented without requiring custom GPU kernels. it is essentially a smarter checkpointing tailored to LLMs training, which takes advantage of the causal nature of the transformers architecture and divides the computation over the time dimension into chunks which are computed sequentially in order to save memory.

S2 The method shows great advantage in memory and speed compared to checkpointing and a competition method named MsT (see end of my summary). It is quite remarkable that such a gain can be obtained without using custom CUDA or triton kernels, but just by cleverly chunking the input sequence.

S3 The method is tailored for long sequence post-training scenarios (SFT, DPO and GRPO) that use long sequences, which is timely given the recent rise of reasoning models using chains of thought and thus generating very long sequences.

**Weaknesses**

W1 An experiment where the authors train a model with SFT, DPO, or GRPO is missing. Having this experiment would give additional strength to the proposed approach, since even though in infinite precision the derivative is computed in the same way as in standard back-propagation or checkpointing, numerical errors might make it more or less accurate to use the proposed approach. The experiment could also  show if this method enables post-training with long sequences on limited hardware and in a reasonable time.

W2 Exposition can be improved. I feel that the exposition of the idea when applied to the last layer outputting the logits in the case of SFT, DPO and GRPO to be given too much spotlight, since the applications to those cases is relatively straightforward from the main idea. I would have liked more focus on the attention part which is less straightforward and it’s where the time saving comes (See also the questions section).

W3 Limited novelty. The approach can be seen as a direct follow up of MsT [1] where the chunking of the input sequence is also proposed but is done at a lower granularity. This is a minor weakness though since the improvements are still remarkable even when compared to MsT.

**References:**
[1] Cheng Luo, Jiawei Zhao, Zhuoming Chen, Beidi Chen, and Anima Anandkumar. Mini-sequence transformer: Optimizing intermediate memory for long sequences training. arXiv preprint arXiv:2407.15892, 2024.

---

> ### Author Rebuttal · Authors · 2025-07-30
>
> We thank the reviewer for the valuable comments. We address all your concerns below.
>
> **Reduction in FLOPS (Q1).** The FLOPs reduction is due to that StreamBP avoids computing the most of the upper triangular entries of the $QK^{\top}$ matrix, which are not used in the attention computation due to the causal structure of language models. Specifically, the attention implementation is based on PyTorch's `scaled_dot_product_attention` with backend being flash attention. Based on the official document, the full $QK^{\top}$ matrix will be computed as long as custom attention mask is required, which is mandatory when using techniques such as padding or sliding window attention. When using StreamBP, since the forward is partitioned along sequence dimension, we are able to perform fine-grained control of the attention computation. Specifically, for the sequence partition $(t_1, t_2)$, we compute `Q[t1:t2, :] K[:t2, :].transponse()` rather than `Q[t1:t2, :] K[:, :].transponse()` for avoiding unnecessary computations. The computation is based on flash attention without materializing the partitioned pre-attention score in the memory. We remark that StreamBP offers flexibility for fine-grained control of attention over sequence level without the need to implement a customized kernel. We will provide more discussions on the reduced FLOPs in the next version of our manuscript.
>
> **Numerical stability of StreamBP in model training on SFT/GRPO/DPO (W1).** Thank you for the suggestion. We have verified numerical accuracy of StreamBP in Table 6, where StreamBP obtains the gradient that has very close precision error as standard BP. In the following tables, we present the evaluation loss of training Qwen 3-4B model for 500 steps with SFT, DPO, and GRPO, respectively.
>
> | Step | Base   | StreamBP | Difference |
> |:----:|:------:|:--------:|:----------:|
> | 1    | 5.7821 | 5.7821   | +0.0000    |
> | 100  | 4.9235 | 4.9256   | +0.0021    |
> | 200  | 4.3128 | 4.3094   | -0.0034    |
> | 300  | 3.9896 | 3.9929   | +0.0033    |
> | 400  | 3.7914 | 3.7947   | +0.0033    |
> | 500  | 3.6670 | 3.6631   | -0.0039    |
>
> **Table:** Evaluation loss of SFT on dataset Capybara.
>
> | Step | Base Loss | StreamBP Loss | Difference |
> |:----:|:---------:|:-------------:|:----------:|
> | 1    | 1.7023    | 1.7023        | +0.0000    |
> | 100  | 1.0874    | 1.0895        | +0.0021    |
> | 200  | 0.9378    | 0.9346        | -0.0032    |
> | 300  | 0.8759    | 0.8792        | +0.0033    |
> | 400  | 0.8360    | 0.8399        | +0.0039    |
> | 500  | 0.8007    | 0.7963        | -0.0044    |
>
> **Table:** Evaluation loss of DPO on dataset Ultrafeedback.
>
> | Step | Base Loss | StreamBP Loss | Difference  |
> |:----:|:---------:|:-------------:|:----------------------------:|
> | 1 | 0.0127 | 0.0127 |  +0.0000  |
> | 100 | 0.0110 | 0.0108 |  -0.0002  |
> | 200 | 0.0048 | 0.0044 |  -0.0004  |
> | 300 | 0.0056 | 0.0056 |  0.0000  |
> | 400 | 0.0048 | 0.0049 |  +0.0001  |
> | 500 | 0.0043 | 0.0042 |  -0.0001  |
>
> **Table:** Evaluation loss of GRPO on dataset tldr.
>
> One can see that the loss of StreamBP is very close to that of the standard BP. The tiny difference is due to different order of float operations; see Appendix B.1 for more details. We will add this additional verification experiment in the next version.
>
> **Exposition can be improved (W2 & Q2).** Thank you for the suggestion. We plan to move the derivation of the GRPO loss to the appendix, as it becomes similar to that of the SFT loss once we observe that the GRPO loss is linearly decomposable along the token dimension. However, we would like to retain the derivation of the DPO loss in the main text, as it involves nontrivial observations of the gradient operator and requires a more careful decomposition. With the space saved, we intend to provide additional details on the attention part as suggested by the reviewer.
>
> **Limited novelty; can be seen as a direct follow up of MsT (W3).** Though it seems a little arguing as the reviewer mentioned this is a minor weakness, we would like to briefly clarify the fundamental differences betwee MsT and StreamBP. 1) MsT sequentially forwards all partitioned inputs before the full backward on the output, while StreamBP performs backward on the partitioned output immediately after each partitioned forward. This seemingly subtle difference results in large memory gap in terms of saving the layer activation. 2) Fundamentally different from the design idea of StreamBP (i.e., linear decomposition of the gradient computation; see our equation (1)), MsT's main design relies on a linear loss decomposition of different partitions (i.e., $L = L_1+\ldots+L_D$), which is not applicable when the loss is nonlinear across sequence dimension, e.g., the DPO objective. 3) StreamBP involves nontrivial partitions of the attention computation, while this part is not present in MsT. In summary, though these two approaches share certain similarities in terms of sequence partition, we believe that StreamBP is fundamentally different from MsT. For more details about the difference, we refer to our response to Reviewer 8uED.
>
> **Lines 58-59, “nontrivial and intricate …” (another Q2).** We apologize for any possible confusion. We meant to express that though the main idea of linearly decomposing the gradient computation seems simple, applying it to the concrete transformer (last layer, MLP, and attention) requires nontrivial and intricate developments, especially in the attention computation. We will clarify this part in the next version.
>
> **Peak memory appears on  at the end of the second forward pass, not the first (Q3).** The additional memory cost at the end of the second forward pass is the model gradient, which is accumulated during the first backward pass (annotated in Figure 1). Please refer to Appendix C.1 for a detailed explanation of the memory profile.
>
> **Minor:**
> * **M1.** Yes, the batch size is 1. We will add the information in the caption.
> * **M2.** It should be $S^{(i)}$, which is the pre-attention score.
> * **M3.** $L$ can be arbitrary objective that depends on the output $Z$. We will provide clarification in the manuscript.
>
> We hope that our response is satisfactory and addresses all your concerns. If there are any additional questions and concerns, please feel free to let us know during the author-reviewer discussion period. We are more than happy to provide further clarifications.

---

> > ### Comment · Reviewer_RVuF · 2025-08-01
> >
> > Thank you for addressing my concerns. In particular for clarifying the point regarding FLOPs and conducting additional robustness tests by fine-tuning a model. I am therefore raising the score from 4 to 5.

---

> > > ### Author Response · Authors · 2025-08-02
> > > **Thank you for raising the score**
> > >
> > > We thank the reviewer for carefully reading our response and raising the score. In the next version of our manuscript, we would include more detailed discussion on the reduced FLOPs of StreamBP and provide comprehensive experiments on training LLMs using StreamBP under different objectives. We would also adjust the writing in Section 3 and allocate more space for explaining StreamBP on attention module. Furthermore, an explicit clarifications on the difference between StreamBP and MsT woud be included.
> > >
> > > Again, thank you for your active participation in the rebuttal procedure of our manuscript!

---

### Official Review · Reviewer_8uED · 2025-06-05

**Clarity:** 4
**Significance:** 3
**Originality:** 2
**Rating:** 4
**Confidence:** 3

**Summary:**

Training large language models on long sequences demands substantial GPU memory during backpropagation. To address this, this paper introduces ​StreamBP, a memory-efficient and exact backpropagation method that processes sequences in chunks and sequentially accumulates gradients across them. By leveraging the causality of self-attention, StreamBP further reduces FLOPs during backpropagation by eliminating redundant attention computations. Compared to gradient checkpointing, StreamBP achieves faster speed and better memory efficiency during backpropagation.

**Questions:**

1. Could you elaborate the key difference between MsT and StreamBP?
2. Line 50, why MsT does not apply to RL objectives?

**Ethical Concerns:**

["NO or VERY MINOR ethics concerns only"]

**Final Justification:**

Based on the author's response, I have re-evaluated the novelty of this paper and raised the rating to Borderline accept.

**Limitations:**

yes

**Quality:**

3

**Strengths And Weaknesses:**

Strengths
1. The paper is clearly written and well motivated. Although this is not my area of expertise, the diagrams help me quickly get up to understand the problem and follow the proposed solution.
2. Compared to gradient checkpointing, StreamBP achieves faster speed and better memory efficiency.
3. StreamBP is applicable to common scenarios such as SFT, GRPO, and DPO.

Weaknesses
1. The novelty is questionable, as the core idea of sequence partition already exists in MsT and Sequence parallel.
2. Table 3 lacks the comparison between MsT and StreamBP, making it unclear whether StreamBP is more memory-efficient than StreamBP.

---

> ### Author Rebuttal · Authors · 2025-07-28
>
> We thank the reviewer for the valuable comments. We now clarify the key differences between StreamBP and the baseline approaches.
>
> **Difference with MsT.** While MsT and StreamBP share certain similarities in terms of sequence partition, they exhibit fundamental differences that result in large efficiency gap and different application scope.
>
> 1. Fundamental difference of the BP order that leads to large memory gap. MsT sequentially forwards all partitioned inputs *before* the full backward on the output (see their Algorithm 1). In comparison, StreamBP performs backward on the partitioned output *immediately* after each partitioned forward. This seemingly subtle difference is fundamental, as MsT still stores the full layer activation while StreamBP only stores the partitioned layer activation. This is because the activation value will only be released from memory after it has been utilized for gradient calculation. Therefore, the main memory saving of MsT comes from the reduced logits in the language modeling head layer, which is tailored to the SFT objective as we will analyze below.
>
> 2. MsT is tailored to SFT; the technique cannot be applied to RLHF objectives, e.g., DPO. MsT’s forward (see their Algorithm 2) and backward (see their Algorithm 4) process of the language modeling head are specifically tailored to SFT objective. Specifically, MsT backwards the loss of one sequence partition at a time for avoiding storing the full logits. However, the technique is not applicable to RLHF objectives such as DPO, due to the linear loss decomposition idea of MsT (i.e., $L = L_1+\ldots+L_D$) and the fact that the nonlinear sigmoid loss of DPO is not linearly decomposable across the sequence dimension. By contrast, StreamBP has a fundamentally different design idea, i.e., linear decomposition of the gradient computation; see our equation (1). This main design idea of StreamBP allows to derive specific forward and backward processes for RLHF objectives, which involves gradient analysis of nonlinear operator.
>
> 3. StreamBP introduces nontrivial partitioning of attention computation, which is not present in MsT. This design in attention part improves both memory and computation efficiency of StreamBP. In particular, MsT adopts the standard attention implementation, which stores the full activations in attention layer including full $Q$, $K$, $V$ matrices. In comparison, StreamBP partitions the attention computation along the sequence dimension (see our Figure 2 and Equation 13) and reduces the activation values by approximately $D$ times, where $D$ is the number of partitions. This partition design is nontrivial due to the causal feature of the attention operator, where the output at position $t$ depends not only on the current position of input, but also on all the input at previous positions. Moreover, StreamBP reduces the computational FLOPs by leveraging the causal structure of language models. Namely, it avoids computing the key and value behind the current partition position. As shown in Table 1, the fine-grained attention computation allows StreamBP to achieve remarkable acceleration over MsT and even the vanilla gradient checkpointing method.
>
> 4. StreamBP proposes a general principle for reducing the memory cost of training neural networks. Unlike MsT that uses loss decomposition (i.e., $L = L_1+\ldots+L_D$), we would like to highlight that StreamBP’s main design idea of decomposing Jacobian-vector product (see our equation (1)) is applicable to general neural network training. For any transformation inside the model, if computing one partition of the output requires storing much less activation values than that for computing full output, then one can apply StreamBP to greatly reduce the memory cost.
>
> **Difference between StreamBP and sequence parallel.** Sequence parallel (SP) is specifically designed for distributed training with multiple available GPUs, while StreamBP is applicable for both single GPU and multi-GPUs settings. Specifically, SP partitions the activation across different GPUs, where each GPU receives one activation partition evaluated on the same input data. Consequently, SP requires a sufficient number of GPUs to scale to long sequences. Additionally, as different segments of the activation are stored in different GPUs, SP requires additional communication of activation values between multiple GPUs. By contrast, StreamBP sequentially processes all partitions on the same GPU, storing only one partition at a time. This design allows StreamBP to facilitate long sequence training even in *a single GPU* setting, while also allowing further scaling of sequence length when multiple GPUs are available.
>
> In summary, StreamBP differs fundamentally from both MsT and sequence parallel in algorithmic design, memory efficiency, and applicable scenarios. These distinctions support our claim of StreamBP’s novelty.
>
> **"Table 3 lacks the comparison between MsT and StreamBP".** We present the memory comparison between StreamBP and MsT in Figure 4 rather than Table 3, as MsT is specifically designed for SFT as outlined above. When the sequence length scales up, there is a clear gap in memory consumption between StreamBP and MsT. The memory gap is caused by layer activation values, as we discussed above (Difference with MsT, 1&3).
>
> **"MsT does not apply to RL objectives".** Please see the discussion above (Difference with MsT, 2).
>
> We hope that our response is satisfactory and addresses all your concerns. Should you have any further questions and concerns, please feel free to let us know during the author-reviewer discussion period. We are more than happy to provide further clarifications.

---

> > ### Comment · Reviewer_8uED · 2025-08-01
> >
> > Thank you for the response. I have re-evaluated the novelty and raised the rating to Borderline accept.

---

> ### Author Response · Authors · 2025-08-01
> **Thank you for raising the score**
>
> Thank you for recognizing the novelty of StreamBP and for increasing your score. In the next version of our manuscript, we will clarify the differences between StreamBP and MsT, particularly in terms of the backpropagation order, applicable training objectives, and attention computation partitioning. We will also elaborate on the distinctions between StreamBP and sequence parallelism, including their respective design choices and GPU requirements.

---

### Official Review · Reviewer_ub3z · 2025-06-30

**Clarity:** 3
**Significance:** 4
**Originality:** 3
**Rating:** 5
**Confidence:** 2

**Summary:**

This paper is about a technique to reduce memory footprint when gradient checkpointing is used.
The main idea is to split values in each feedforward layer into chunks, so that the entire values do not need to be recomputed in the re-forward step of gradient checkpointing.
The main target for the reduction technique suggested is the Transformer architecture, and various splitting strategies are suggested for each component of the architecture, including objective functions from SFT, GRPO, DPO and Transformer componenets like attention, MLP.
Due to the splitting strategies adapted to each component, it can benefit in computation efficiency due to reduced sizes of matrices used in computing causal operators like attention.
From evaluations, it is empirically proven that in terms of memory cost the StreamBP method outperforms baselines without checkpointing by 23-36x and those with checkpointing by 2.8-5.5x.
Also in terms of time cost, the proposed method generally outperformed the plain checkpointing technique, especially when the length of context gets longer (likely due to reduced computation in attention).

**Questions:**

* To my assumption in exchange of reduced memory cost the StreamBP method needs to do computation multiple times sequentially, i.e. for the number of chunks/blocks. However actually it resulted in less time required for training, is it solely due to partitioning of attention layers while considering causal relationship? If so, I want to see the speed degradation in other layers vs. speed improvement in attention layers.

**Ethical Concerns:**

["NO or VERY MINOR ethics concerns only"]

**Final Justification:**

My concerns are resolved in the author response and I changed the score. Thank authors for the clarification.

**Limitations:**

It would be more nice to mention if there are cases that this technique cannot be used.

**Quality:**

3

**Strengths And Weaknesses:**

Strengths
* Good overall performance improvement
* Even with more fine-grained checkpointing, it is actually performant in terms of time cost.

Weaknesses
* Comparison with possibly similar prior work like MoNET, Block Parallel Transformer would make the statement stronger.

---

> ### Author Rebuttal · Authors · 2025-07-30
>
> We thank the reviewer for the valuable comments. We now address your concerns and questions below.
>
> **Comparison with MoNET [1]**
> * StreamBP and MoNET are designed for different tasks. In particular, MoNET is primarily used for vision tasks, and is specifically optimized for convolution-heavy operators, e.g., batch normalization and max-pooling. In comparison, StreamBP focuses on reducing the activation memory cost of the training of transformer LLMs, where the activation memory can usually be much higher.
>
> * The core methodology between StreamBP and MoNET are essentially different. MoNET dynamically determines which quantities to checkpoint for minimizing the memory consumption. By contrast, StreamBP adopts the layer-wise checkpointing strategy and uses partitioned forward and backward of a single layer for reducing the memory cost.
>
> **Comparison with Block Parallel Transformer (BPT) [2]**
>
> * BPT only reduces the activation memory in transformer layer, while StreamBP is applicable to general mapping, e.g., language modeling head, which further reduces the memory consumption by a large margin. By applying StreamBP on the language modeling head, the logits memory is significantly reduced by more than 90%; see Figure 8. In certain setting (such as Figure 1), the memory consumption of the language modeling head can be huge, and hence reducing the memory cost of this part could be important.
> * StreamBP reduces computational FLOPs by leveraging the causal structure of LLM. In particular, it saves approximately half computation of the pre-attention score; see lines 188-195. In comparison, BPT still computes the full pre-attention score.
> * StreamBP is compatible with most of the fused attention kernels, e.g., flash attention, and thereby has much less HBM throughput than BPT.
>
> We note that BPT is implemented using JAX. It is highly nontrivial to reproduce the exact BPT implementation in PyTorch for an experiment comparison. We will provide further discussions on MoNET and BPT in the next version of our manuscript.
>
> **Speed degradation of applying StreamBP in layers other than attention layer.** The BP time acceleration of StreamBP comes from the reduced computational FLOPs in attention score calculation and the reduced HBM throughput of attention mask. To see StreamBP's effect on BP speed degradation, we apply StreamBP on the language modeling head only, and compare it to the gradient checkpointing baseline. We perform 2 rounds of BP on a single A800 GPU and summarize the results in the following table:
>
> | Partition   Size   | 100   | 200   | 500   | 1000  | 3000  | \| Gradient Checkpointing |
> |------|-------|-------|-------|-------|-------|---------------------------|
> | Memory (GB)    | 18.82 | 18.85 | 18.94 | 19.08 | 22.77 | \| 34.76                  |
> | Time (s) | 5.15  | 5.03  | 4.96  | 4.92  | 4.89  | \| 4.83                   |
>
> **Table:** BP memory and time cost of Qwen 3-4B under sequence length 9k.
>
> From the above table, StreamBP introduces slight overhead compared to gradient checkpointing method, even though the total computational FLOPs is the same. The overhead is due to repetitive kernel launch and increased HBM throughput. As the partition size increases, the number of partitions decreases, leading to reduced BP time.
>
> We hope that our response is satisfactory and addresses all your concerns. If there are any additional questions and concerns, please feel free to let us know during the author-reviewer discussion period. We are more than happy to provide further clarifications.
>
> **References**
>
> [1] Aashaka et al, "Memory Optimization for Deep Networks", ICLR 2021.
>
> [2] Liu et al, "Blockwise Parallel Transformer for Large Context Models", NeurIPS 2023.

---

### Official Review · Reviewer_ucaR · 2025-07-02

**Clarity:** 3
**Significance:** 2
**Originality:** 3
**Rating:** 5
**Confidence:** 3

**Summary:**

This paper introduces StreamBP, a memory-efficient and exact back-propagation method for long-sequence training of LLMs. StreamBP uses layer-wise linear decomposition of the chain rule along the sequence, resulting in reduced memory cost of activation and logits. Compared to gradient checkpointing, StreamBP is able to support 2.8$\times$ longer maximum sequence length.

**Questions:**

As listed in Weaknesses above. It would be great if the authors could elaborate on the above 3 points.

**Ethical Concerns:**

["NO or VERY MINOR ethics concerns only"]

**Final Justification:**

My concerns have been sufficiently addressed in authors' rebuttal.

**Limitations:**

The current evaluation results are presented without any variance metrics (e.g. across seeds or runs), which weakens the confidence in the claim that StreamBP achieves.

**Quality:**

3

**Strengths And Weaknesses:**

Strengths:

- The paper intends to tackle a timely and important topic of reducing memory usage for long-sequence training of LLM;
- The proposed method of streaming partial gradients to avoid storing full activations while keeping gradients exact (instead of approximating) seems novel and effective;
- The empirical results demonstrate reduced activation memory usage on standard language modelling objectives e.g. SFT, DPO, and GDPO.

Weaknesses:

- Optimizing memory consumption for activation and logits is important, and the paper show effectiveness of streamBP towards this objective. However, in practice, it is not sufficient to consider only activation and logits for scaling large models and long sequences. As shown Fig. 3, model weights takes significant memory especially when model grows bigger. This should be explicitly discussed. One suggestion would be to discuss how StreamBP can complement weight, gradient, optimiser memory reduction methods such as GaLore and BAdam, to address the full memory bottleneck in large-model, long-context training.
- The authors mentioned sequence parallel approaches increase the maximum sequence length, but there lacks further discussion or comparison with such methods. A short discussion of trade-offs and complementarity would strengthen the paper and help practitioners understand when StreamBP is preferable or combinable with existing parallelism strategies.
- Current evaluation is on Qwen only, it would be good to see results with other SOTA LLMs to confirm StreamBP generalise well with different architectures.

---

> ### Author Rebuttal · Authors · 2025-07-30
>
> We thank the reviewer for the valuable comments. We now address your concerns below.
>
> **How StreamBP complements existing memory-efficient methods for training LLM.** Thank you for the suggestion. The memory cost of LLM training mainly consists of storing activation values, model weight, and optimizer states (including model gradient). StreamBP reduces the memory cost of activation values during BP, and can be directly combined with existing memory-efficient training methods that reduce model memory (e.g., quantization methods, QLoRA) and optimizer states memory (e.g., LoRA, Galore, BAdam) for further reducing the memory bottleneck caused by model weight and optimizer states, respectively. Since StreamBP is an exact BP approach, combining it with other memory-efficient methods will not result in any performance degradation of the method. In our manuscript, we have provided training results for combining StreamBP and LoRA in Table 3. We will explicitly state that StreamBP can be directly applied to existing memory-efficient methods such as QLoRA, LoRA, Galore, BAdam in the next version of our manuscript.
>
> **Comparison between StreamBP and sequence parallel.** Sequence parallel (SP) is specifically designed for distributed training with multiple available GPUs, while StreamBP is applicable for both single GPU and multi-GPUs settings. Specifically, SP partitions the activation across different GPUs, where each GPU receives one activation partition evaluated on the same input data. Consequently, SP requires a sufficient number of GPUs to scale to long sequences. Additionally, as different segments of the activation are stored in different GPUs, SP requires additional communication overheads between multiple GPUs. By contrast, StreamBP sequentially processes all partitions on the same GPU, storing only one partition at a time. This design allows StreamBP to facilitate long sequence training even in *a single GPU* setting, while also allowing further scaling of sequence length when multiple GPUs are available. We believe that this clearly clarifies the fundamental difference between StreamBP and SP. In the next version, we will add this discussion about these two approaches.
>
> **Evaluation on different model architectures.** We thank the reviewer for the suggestion. As StreamBP is an algorithm that linearly decomposes the chain rule during gradient computation in a transformer architecture, it is not tailored to any specific model architectures. Hence, it can be applied to any transformer LLMs. In the following tables, we present the BP memory cost of Llama and Gemma models using StreamBP and standard BP.
>
> | Sequence Length     |  3k    |   6k    |   9k    |  18k    |
> |---------------------|--------|---------|---------|---------|
> | StreamBP            | 31.81  | 32.66   | 33.62   | 37.20   |
> | Base w/ ckpt        | 35.79  | 41.69   | 47.61   | 65.59   |
> | Base w/o ckpt       | 54.20  | -       | -       | -       |
>
> **Table:** BP memory cost (GB) of Llama 3.1-8B.
>
> | Sequence Length     |  3k    |   6k    |   9k    |  18k    |
> |---------------------|--------|---------|---------|---------|
> | StreamBP            | 4.69   | 4.95    | 5.37    | 7.85    |
> | Base w/ ckpt        | 14.28  | 24.79   | 35.33   | 67.16   |
> | Base w/o ckpt       | 23.08  | 42.80   | 63.37   | -       |
>
> **Table:** BP memory cost (GB) of Gemma 3-1B.
>
> Clearly, the memory cost of StreamBP is significantly less than standard BP; the memory gap increases as the sequence length scales up, which verifies the efficiency of StreamBP on different model architectures. Notably, Gemma 3-1B costs even more memory than Llama 3.1-8B for standard BP under sequence length 18k. This is because Gemma 3 model has 2$\times$ larger vocabulary size than Llama 3.1, and hence doubles the memory cost of logits. In comparison, StreamBP only stores partitioned logits and its memory cost is dominated by the model weight and model gradient rather than activation values.
>
> **The evaluation results are presented without variance metrics.** Given the fixed model, sequence length, and partition size, the memory cost of StreamBP will be a constant and does not vary across different seeds and runs.
>
> We hope that our response is satisfactory and addresses all your concerns. If there are any additional questions and concerns, please feel free to let us know during the author-reviewer discussion period. We are more than happy to provide further clarifications.

---

> > ### Comment · Reviewer_ucaR · 2025-08-01
> >
> > Dear authors of StreamBP,
> >
> > Thank you for your response to my review comments. My concerns have been sufficiently addressed, and I am raising my score to Accept.
> >
> > Best of luck!

---

> > > ### Author Response · Authors · 2025-08-02
> > > **Thank you for raising the score**
> > >
> > > We thank the reviewer for carefully reading the response and rasing the score. In the next version of the manuscript, we would compare StreamBP with memory-efficient methods and sequence parallel, and include the evaluation of different architectures. We sincerely appreciate the reviewer's valuable comments. Good day!

---

### Decision · Program_Chairs · 2025-09-17

**Decision:**

Accept (poster)

**Comment:**

On this paper, the authors propose a method to improve the memory efficiency of LLM training, by proposing an update to the backpropagation step. They do so, by first partitioning the input of each layer (e.g., Attenion, output layer, MLP) and feeding them sequentially both during the forward and the backward pass. The main idea is that then the entire values do not need to be recomputed during the gradient checkpointing. The authors show that the computed gradients are exact, and that the method works on different training strategies such as SFT, DPO and GRPO (RL). They also show that by using this method it is possible to go to quite extreme training tokens, in particular, they are able to train a Qwen3-14B with a sequence of 48k tokens in a single 80GB GPU

The paper initially got borderline reviews, 3 Borderline Accepts and  1 Borderline Reject. The reviewers praised the experimental section, its generalization to different tasks, good overall performance and the writing of the paper. However, they were divided into the novelty of the work. At the same time, the reviewers found some weaknesses in the paper when it comes to evaluating only in Qwen (the authors showed in rebuttal evaluation in Gemma), asking for some clarification which the authors properly addressed in the rebuttal, more comparisons with prior works (which the authors did in the rebuttal), clarification in number of flops and asking for experiments in numerical stability of the method in model training on SFT/GRPO/DPO. The authors gave good explanation and responses to all these points in the rebuttal.

After the discussion stage, all the reviewers increased their scores, with the final scores  being 3 Accepts and 1 Borderline Accept, unanimously recommending the paper to be accepted. I agree with the reviewers about the decision and justification, and urge the authors to implement all the changes in the camera-ready version of the paper. Congratulations to the authors for a very nice NeurIPS paper!